# Gathering the Evidence on Diet and Depression: A Protocol for an Umbrella Review and Updated Meta-Analyses

**DOI:** 10.3390/mps6050078

**Published:** 2023-08-31

**Authors:** Alexandra M. Bodnaruc, Coralie Vincent, Carolina Soto, Miryam Duquet, Denis Prud’homme, Isabelle Giroux

**Affiliations:** 1School of Nutrition Sciences, Faculty of Health Sciences, University of Ottawa, Ottawa, ON K1N 6N5, Canada; abodn049@uottawa.ca (A.M.B.); cvinc064@uottawa.ca (C.V.); csoto007@uottawa.ca (C.S.); mduqu016@uottawa.ca (M.D.); 2Université de Moncton, Moncton, NB E1A 3E9, Canada; denis.prudhomme@umoncton.ca

**Keywords:** depression, depressive disorders, mental health, dietary patterns, food groups, foods, nutrients, overview of reviews, systematic review, meta-analysis

## Abstract

Our objectives are to perform (1) an umbrella review on diet and depression, (2) a systematic review update on dietary patterns and depression, and (3) updated meta-analyses using studies from the previous two objectives. Systematic reviews examining the relationships between diet and depression and primary studies on the relationship between dietary patterns and depression will be systematically retrieved via several databases. All articles identified through the database searches will be imported into Covidence. Following duplicates removal, two authors will independently perform title and abstract screening and full-text assessment against eligibility criteria. Data will be extracted using tables developed for both systematic reviews and primary studies. The methodological quality of systematic reviews will be assessed using the AMSTAR-2 tool. The risk of bias in randomized trials, cohort and cross-sectional studies, as well as case-control studies, will be assessed with the Cochrane risk-of-bias (RoB-2) tool, the NHLBI Quality Assessment Tool for Observational Cohort and Cross-Sectional Studies, and the NHLBI Quality Assessment Tool for Case-Control studies, respectively. For each dietary variable, data extracted will be used to produce: (1) a summary of systematic reviews’ characteristics and results table, (2) a summary of the primary studies characteristics table, (3) a qualitative summary of results from the primary studies table, and (4) a quantitative summary of results in the form of forest plots. The certainty of evidence will be assessed using the Grading of Recommendations, Assessment, Development and Evaluations (GRADE) approach. Upon completion, this systematic review will be the most comprehensive and up-to-date synthesis of currently available evidence on the relationships between diet and depression. It will serve as a key reference to guide future research and as a resource for health professionals in the fields of nutrition and psychiatry. PROSPERO CRD42022343253.

## 1. Introduction

Depression is a complex and heterogeneous disorder characterized by a broad set of psychological and physiological symptoms, including persistent low mood, anhedonia, emotional dysregulation, reduced cognitive abilities, fatigue, as well as appetite and sleep disturbances [1]. According to the Global Burden of Diseases, Injuries, and Risk Factors Study 2019, depression is the most widespread mental health disorder, affecting over 250 million individuals of all ages and placed second among the leading causes of disability worldwide [2]. Individuals with depression experience significant impairments in their ability to perform activities of daily living as well as fulfill their social, familial, and professional functions [3,4,5,6], all of which contribute to reducing their quality of life [7] and have critical economic and societal repercussions [8,9]. The burden of depression is further amplified by its high comorbidity rates [10,11,12,13,14,15] and the limited effectiveness of pharmacological treatments [16,17,18,19]. Indeed, depression has multiple psychiatric [13,14,15] and cardiometabolic comorbidities [10,11,12], which generally worsen associated prognostics and increase mortality rates [20,21,22]. Furthermore, currently available anti-depressants, which are a mainstream treatment choice, were shown to significantly reduce (i.e., ≥50% reduction in the severity) depressive symptoms in about two-thirds of patients and to lead to full remission (i.e., disappearance of depression diagnosis criteria for ≥2 months) in only about one-third of them [16,17,18,19].

Over the past 15 years, in an effort to improve prevention strategies and to complement currently available treatments, there has been a growing scientific interest in studying the relationships between diet and depression [23,24,25]. In addition to genetics, which accounts for 30–40% of the risk of developing depression [26,27], environmental factors such as a history of childhood abuse or neglect [28,29], adverse life events [30,31], social isolation [32,33], socioeconomic status [34,35], and lifestyle [36,37], play a crucial role. Diet, described as a modifiable environmental risk factor, may influence the risk of depression and depressive symptoms through its effects on components that are part of depression’s pathophysiological pathways [38], such as monoamines’ metabolism [39,40], neurotrophic factors’ synthesis [41,42], the hypothalamic–pituitary–adrenal axis [43,44], oxidative stress [45,46,47], and inflammation [48,49,50,51]. Evidence from observational and experimental studies suggests significant associations between several dietary components (e.g., fruit and vegetables, omega-3 fatty acids) and depression [52,53,54,55,56,57,58,59,60,61,62,63,64,65,66,67,68,69,70], as well as between diet as a whole (e.g., diet quality, adherence to specific dietary patterns) and depression [71,72,73,74,75,76,77,78,79,80,81,82,83]. While there is evidence to support these associations, there are also multiple inconsistencies that have not been ruled out by systematic reviews and meta-analyses on the topic.

The growing interest in this field of study has contributed to a significant increase in the number of publications of both primary studies and systematic reviews on the topic (see Figure 1A,B). The high rate of systematic review publication, combined with the low rate of prospective protocol registration, has led to the publication of a large number of reviews on almost identical topics covering similar publication periods. For example, in 2019, at least eight systematic reviews have been published on the topic of dietary patterns and depression [73,74,75,76,77,78,79,80], out of which only three had prospective protocol registrations [73,74,77]. This issue has been previously highlighted in several fields of research [84,85,86] and needs to be addressed in order to avoid contrasting and potentially misleading conclusions, as well as to ensure that research efforts are distributed efficiently where most needed.

Based on the findings and issues presented above, our general objectives, illustrated in Figure 2, are the following:(1)To perform an umbrella review on diet and depression. This objective will allow us to synthesize the characteristics, methodology, results, and extent of overlap of existing systematic reviews on diet and depression.(2)To perform a systematic review update on dietary patterns and depression. This objective will allow the identification of more recently published primary studies on dietary patterns and depression that were not included in the latest systematic reviews on the topic. The second objective is limited to dietary patterns due to the extensive nature of this work. The decision to limit the dietary variable to dietary patterns specifically is based on the growing interest in this approach and the recommendation to consider diet as-a-whole rather than individual components of the diet [87,88]. This recommendation is reflected in the increasing number of publications on this specific topic (see Figure 1C).(3)To perform updated meta-analyses using studies from the previous two objectives This objective will allow the provision of the most up-to-date synthesis of the literature on diet and depression.

## 2. Methods

This protocol has been registered within the PROSPERO database for systematic reviews and meta-analyses (registration number: CRD42022343253). A preliminary search of PROSPERO, Cochrane Database of Systematic Reviews and the Joanna Briggs Institute Database of Systematic Reviews and Implementation Reports was conducted, and no existing or ongoing systematic reviews with similar aims were identified. The protocol is reported following the Preferred Reporting Items for Systematic Reviews and Meta-Analyses Protocols (PRISMA-P) statement [89] (Appendix A) and was informed by Lunny et al.’s (2017, 2018) [90,91] framework of methods for conducting, interpreting, and reporting overviews of systematic reviews, as well as Cochrane Guidelines for Systematic Reviews of Interventions [92].

### 2.1. Objective 1–Umbrella Review on Diet and Depression

#### 2.1.1. Identification of Relevant Studies

Systematic reviews documenting the relationships between diet and depression will be systematically retrieved via Medline, EMBASE, PsycINFO, Web of Science, the Cochrane Database of Systematic Reviews, and the Joanna Briggs Institute Database using database-specific subject headings and keywords. The search will be limited to articles published in English or French between 1 January 2005 and the date of the last search. The publication period has been restricted as no systematic reviews on diet and depression were published prior to 2005. Additionally, most systematic reviews published between 2005 and 2010 have been updated at least once. The search strategy for PsycINFO as well as the search strategies for all databases are presented in Table 1 and Appendix A, respectively.

#### 2.1.2. Eligibility Criteria

##### Type of Participants

Systematic reviews focusing on (i) healthy children, adolescents and adults and/or (ii) children, adolescents and adults with diagnoses of primary unipolar depressive disorders will be considered eligible. No restrictions will be applied as to participants’ age, sex, and ethnicity, nor females’ sexual development, gestation, or menopause stages. Reviews focusing exclusively on individuals with chronic health conditions, other than depression, will be excluded.

##### Type of Exposures and Interventions

A wide range of dietary variables, namely nutrients and other food components, foods and food groups, as well as dietary patterns, will be considered as eligible. No restrictions will be applied as to the methods and tools used to assess dietary intakes. Systematic reviews focusing exclusively on (i) biomarkers of dietary intake, (ii) nutrient status, (iii) acute dietary interventions, (iv) dietary intervention targeting weight loss through a reduction of energy intake, but no improvement of diet quality, and (v) herbal supplements will be excluded.

##### Type of Comparators

Systematic reviews, including experimental studies. will be considered eligible if the dietary intervention of interest is compared to (i) a placebo, (ii) another dietary intervention, or (iii) no intervention.

##### Type of Outcomes

Outcomes will be limited to unipolar major and persistent depressive disorder with or without current treatment, as well as to depressive symptoms. No restrictions will be applied as to the methods and tools used to assess depression and depressive symptoms. Systematic reviews focusing exclusively on depressive symptoms as part of other physical (e.g., hypothyroidy, anemia, cardiometabolic disorders, etc.) or mental health disorders (e.g., schizophrenia, eating disorders, personality disorders, declined cognitive functions, etc.) symptomatology will be excluded.

##### Type of Study Designs

Systematic reviews of experimental (i.e., randomized controlled parallel and crossover trials with individual and cluster randomization) and observational (i.e., cohort, case-control, and cross-sectional studies) studies will be considered eligible. Systematic reviews of preclinical trials, case studies, and case series will be excluded. We will also exclude systematic reviews in which (i) only one database was searched, (ii) study selection was not performed in duplicate, and (iii) risk of bias in individual studies was not assessed. If original and updated versions of a systematic review are identified, only the most recent updated version will be retained.

#### 2.1.3. Data Collection

##### Selection of Studies

All records identified through the database search will be imported into a Covidence project folder, and duplicates will be removed. All non-duplicate records identified will be screened against title and abstract by two authors (A.M.B./C.V. and A.M.B./C.S., respectively). Articles deemed eligible based on the title and abstract will undergo full-text review, independently, by the same two authors. Any disagreement between authors will be solved through discussion and, when necessary, a third author’s input (M.D., I.G. or D.P.).

##### Data Extraction and Management

Two authors (A.B. and C.V.) will independently perform data extraction from at least 50% of eligible systematic reviews [93]. Data extracted in duplicate will be compared and Cohen’s kappa coefficients will be computed as an indicator of the level of agreement. If Cohen’s kappa values are ≥0.80, one author will complete the extraction of all remaining systematic reviews. If Cohen’s kappa values are <0.80, data extraction will be completed in duplicate for all eligible systematic reviews [93]. Disagreement between authors performing the data extraction will be solved through discussion and, if needed, the input of a third author.

Data extraction tables were developed for systematic reviews and primary studies. As we expect significant overlaps between systematic reviews, we will extract data from systematic reviews per se as well as primary studies included in the systematic reviews. Data extraction tables will be piloted using a small number of studies identified through the database search and necessary adjustments will be made. If necessary, additional changes will be made during the data extraction process. Data extraction fields for systematic reviews and primary studies are shown in Table 2 and Table 3, respectively.

##### Assessment of Methodological Quality

The methodological quality of the included systematic reviews will be independently assessed by two authors (A.M.B. and C.V., C.S. or M.D.) [94]. The authors’ assessments will be compared, and Cohen’s kappa coefficient will be computed as an indicator of the level of agreement between assessors. If Cohen’s kappa values are ≥0.80, one author will complete the assessment of the methodological quality of all studies [94]. If Cohen’s kappa values are <0.80, the methodological quality assessment of all studies will be completed in duplicate [94]. Any discrepancies between authors’ assessments will be resolved through discussion and, if necessary, the input of a third author (I.G. or D.P.).

The methodological quality of systematic reviews will be assessed with the Assessment of Multiple Systematic Reviews 2 (AMSTAR-2) tool [95], which was shown to be valid, moderately reliable, and applicable to systematic reviews of both interventional and observational studies [96,97]. AMSTAR-2 allows describing systematic reviews as being of high, moderate, low, or critically low quality based on 16 items relating to well-established methodological standards. Of these 16 items, 12 (i.e., items 1, 3, 5, 6, 8, 10–16) have a dichotomous, “yes” or “no”, answer choice, while the remaining 4 (i.e., items 2, 4, 7, and 9) can be answered by “yes”, “partial yes”, or “no”. As recommended by Shea et al. (2017), items 2, 4, 7, 9, 11, 13, and 15 will be considered critically important and will, therefore, have a greater impact on the overall quality judgment [95]. Each “no” answer for items considered critically important will be counted as a critical weakness, and, where applicable, each “partial yes” answer will be counted as a non-critical weakness. The overall quality judgment will be based on the number of critical and non-critical weaknesses identified. Where applicable, the overall quality judgement will be downgraded for multiple non-critical weaknesses. Table 4 further details the criteria that will be used.

### 2.2. Objective 2–Systematic Review Update on Dietary Patterns and Depression

#### 2.2.1. Identification of Relevant Studies

Primary studies documenting the relationship between dietary patterns and depression will be systematically retrieved via Medline, EMBASE, PsycINFO, Web of Science, and the Cochrane Central Register of Controlled Trials using both database-specific subject headings and keywords. The search will be limited to articles published in English or French between 1 January 2018 and the date of the last search. The publication period has been restricted as we only want to identify recent studies that have been published outside of the timeframe of previous reviews on dietary patterns and depression. Several systematic reviews on the topic have identified studies published up to mid-2018. The search strategy for PsycINFO as well as the search strategies for all databases are presented in Table 5 and Appendix A, respectively.

#### 2.2.2. Eligibility Criteria

##### Type of Participants

Primary studies focusing on (i) healthy children, adolescents and adults and/or (ii) children, adolescents and adults with diagnoses of primary unipolar depressive disorders will be considered eligible. No restrictions will be applied as to participants’ age, sex, and ethnicity, nor females’ sexual development, gestation, or menopause stages. Primary studies focusing exclusively on individuals with chronic health conditions, other than depression, will be excluded.

##### Type of Exposures and Interventions

Only studies focusing on dietary patterns will be considered eligible. No restrictions will be applied as to the methods and tools used to assess adherence to dietary patterns. Primary studies focusing exclusively on (i) foods and food groups, (ii) nutrients and other food components, (iii) biomarkers of dietary intake, (iv) nutrient status, (v) acute dietary interventions, (vi) dietary intervention targeting weight loss through a reduction of energy intake, but no improvement of diet quality, and (vii) herbal supplements will be excluded.

##### Type of Comparators

Primary experimental studies will be considered eligible if the dietary intervention of interest is compared to (i) a placebo, (ii) another dietary intervention, or (iii) no intervention.

##### Type of Outcomes

Outcomes will be limited to unipolar major and persistent depressive disorder with or without current treatment, as well as to depressive symptoms. No restrictions will be applied as to the methods and tools used to assess depression and depressive symptoms. Systematic reviews focusing exclusively on depressive symptoms as part of other physical (e.g., hypothyroidy, anemia, cardiometabolic disorders, etc.) or mental health disorders (e.g., schizophrenia, eating disorders, personality disorders, declined cognitive functions, etc.) symptomatology will be excluded.

##### Type of Study Designs

Primary experimental (i.e., randomized controlled parallel and crossover trials with individual and cluster randomization) and observational (i.e., cohort, case-control, and cross-sectional studies) studies will be considered eligible. Preclinical trials, case studies, and case series will be excluded.

#### 2.2.3. Data Collection

##### Selection of Studies

All records identified through the database search will be imported into a Covidence project folder, and duplicates will be removed. All non-duplicate records will be screened against title and abstract by two authors (A.M.B./C.V. and A.M.B./C.S., respectively). Articles deemed eligible based on the title and abstract will undergo full-text review, independently, by the same two authors. Any disagreement between authors will be solved through discussion and, when necessary, a third author’s input (M.D., I.G. or D.P.).

##### Data Extraction and Management

Two authors (A.B. and C.V., C.S. or M.D.) will independently perform data extraction from at least 50% of eligible primary studies [93]. Data extracted in duplicate will be compared and Cohen’s kappa coefficients will be computed as an indicator of the level of agreement. If Cohen’s kappa values are ≥0.80, one author will complete the extraction of all remaining primary studies. If Cohen’s kappa values are <0.80, data extraction will be completed in duplicate for all eligible primary studies [93]. Disagreement between authors performing the data extraction will be solved through discussion and, if needed, the input of a third author.

Data extraction tables were developed for primary studies. Data extraction tables will be piloted using a small number of studies identified through the database search and necessary adjustments will be done. If necessary, additional changes will be done during the data extraction process. Data extraction fields for primary studies are shown in Table 3.

##### Assessment of the Risk of Bias

The risk of bias of 50% of the included primary studies will be independently assessed by two authors (A.M.B. and C.V., C.S. or M.D.) [94]. The authors’ assessments will be compared, and Cohen’s kappa coefficient will be computed as an indicator of the level of agreement between assessors. If Cohen’s kappa values are ≥0.80, one author will complete the assessment of the risk of bias of all studies [94]. If Cohen’s kappa values are <0.80, the risk of bias assessment of all studies will be completed in duplicate [94]. Any discrepancies between authors’ assessments will be resolved through discussion and, if necessary, the input of a third author (I.G. or D.P.).

The risk of bias in randomized trials, cohort and cross-sectional studies, as well as case-control studies, will be assessed using the revised Cochrane risk-of-bias (RoB-2) tool [94], the National Heart, Lung, and Blood Institute (NHLBI) Quality Assessment Tool for Observational Cohort and Cross-Sectional Studies [98], and the NHLBI Quality Assessment Tool for Case-Control studies [99], respectively.

The Cochrane RoB-2 tool assesses five bias domains known to affect the results of randomized and quasi-randomized trials, namely (i) bias arising from the randomization process, (ii) bias due to deviations from intended interventions, (iii) bias due to missing outcome data, (iv) bias in the measurement of the outcomes, and (v) bias in the selection of the reported results. Each domain contains guiding questions, which can be answered by “yes”, “probably yes”, “no”, “probably no”, or “no information”. Based on assessors’ answers to these questions, an overall judgement of either low risk of bias, some concerns, or “high risk of bias”, will be reached for each domain. Using judgements reached for each domain, the study itself will be rated as:(i).being at *low risk of bias* when all domains are rated as such,(ii).raising *some concerns* when at least one domain is rated as such, but no domain is rated as being at high risk of bias, and as(iii).being at *high risk of bias* when at least one domain is rated as such, or when multiple domains are rated as raising some concerns.

The NHLBI Quality Assessment Tool for Observational Cohort and Cross-Sectional Studies and the NHLBI Quality Assessment Tool for Case-Control Studies consist of 14 and 12 items, respectively, assessing common sources of bias in observational studies, namely (i) bias from participants’ recruitment or selection methods and sample size, (ii) bias in the measurement of the exposures, (iii) bias in the measurement of the outcomes, (iv) bias due to the handling of potential confounders, and (v) bias in the selection of the reported results. Each item of the NHLBI Quality Assessment Tools can be answered by “yes”, “no”, or “no information”. Cohort and cross-sectional studies will be considered as being at low risk of bias when the answer to ≥13 items is yes, at moderate risk of bias when the answer to 10, 11 or 12 items is yes, and at high risk of bias when the answer to <10 items is yes. Case-control studies will be considered at low risk of bias when the answer to ≥11 items is yes, at moderate risk of bias when the answer to 8, 9 or 10 items is yes, and at high risk of bias when the answer to <8 items is yes.

### 2.3. Objective 3–Updated Meta-Analyses

#### 2.3.1. Data Analysis and Synthesis

##### Outcome Measures

Primary studies included in previous systematic reviews and those identified through the updated systematic review are likely to report continuous, ordinal, and dichotomous outcome data. All types of outcome data will be included in the qualitative synthesis. Continuous data reported as mean differences (with standard deviations (SD)) as well as dichotomous data reported as risk or hazard or odds ratios (with 95% confidence interval (CI)) will be included in the quantitative synthesis of results. When possible, ordinal data will be dichotomized and included in the quantitative synthesis. Missing outcome data will first be dealt with by contacting the authors of the primary publication. If no response is received after two attempts, data imputation will be used and considered in sensitivity analyses.

##### Synthesis of Results

For each dietary variable, data extracted will be used to produce (1) a summary of systematic reviews’ characteristics and results table, (2) a summary of primary studies characteristics table, (3) a qualitative summary of results from primary studies table, and (4) a quantitative summary of results in the form of forest plots. Rather than performing meta-analyses of systematic reviews, we will perform meta-analyses including individual studies included in systematic reviews, combined with studies identified from the updated systematic review on dietary patterns. Pooling the results from systematic review would introduce significant bias, as primary studies included in multiple reviews will have greater statistical power and likely result in overly precise misleading estimates [90,91,100].

##### Qualitative Synthesis of Results

Results from each study, including those that cannot be included in the quantitative synthesis of results, will be categorized as (1) non-significant negative relationship or effect, (2) significant negative relationship or effect, (3) non-significant positive relationship or effect, and (4) significant positive relationship or effect [101]. For each dietary variable, we will create a qualitative summary table including studies’ main characteristics (i.e., year, study design, number of participants, population, and RoB) and results categories, as described above.

##### Quantitative Synthesis of Result

All statistical analyses described below will be carried out using R statistical software [102] with meta [103,104] and metafor [105] packages.


Observational Studies


We will estimate risk ratios (with 95% CI) using the inverse variance method with random-effect models [106]. Risk ratios, odds ratios and hazard ratios will be pooled together. The random-effect model, which considers both within-study and between-study heterogeneity, is selected a priori as we expect non-negligible between-study heterogeneity due to population characteristics as well as the tool used to assess dietary intakes, dietary patterns, depression, and depressive symptoms.


Experimental Studies


We will estimate standard mean differences (with 95% CI) using the inverse variance method with random-effect models [106]. The random-effect model, which considers both within-study and between-study heterogeneity, is selected a priori as we expect non-negligible between-study heterogeneity due to population characteristics, interventions’ characteristics, as well as the tools used to assess dietary intakes, dietary patterns, depression and depressive symptoms.


Subgroup and Sensitivity Analyses


Where applicable, subgroup analyses will be performed for variables such as study characteristics, assessment of exposures, interventions characteristics, assessment of the outcome, risk of bias, and statistical analyses. Sensitivity analyses will be performed by repeating analyses while excluding observational studies with <500 participants, observation, studies in which statistical analyses were not adjusted for confounders, studies including participants <18 years and/or >65 years, studies focusing on women in the peri- and/or post-partum period, studies using non-validated exposure assessment tools, studies using non-validated outcome assessment tools, and studies judged to be at a high risk of bias. Additionally, analyses will be repeated using fixed-effect models.


Assessment of Heterogeneity


The heterogeneity of studies included in each updated meta-analysis will be assessed using both Cochran’s Q [107,108] and I^2^ tests [108,109]. Cochran’s Q value is obtained by summing the squared deviations of each study’s estimate from the meta-analysis pooled estimate, with primary studies’ contributions being weighted identically to the meta-analysis [107]. The associated *p*-value will be obtained using a chi^2^ distribution table, with degrees of freedom corresponding to the number of studies included in the meta-analysis minus one [107]. *p*-values ≤0.10 will be considered as statistically significant and will, thus, be interpreted as indicative of between-study heterogeneity that is not accountable by chance alone [17,108]. The I^2^ test value corresponds to the percentage ratio of between-study variance over the sum of between- and within-study variances [109]. Values ranging between 0 and 40%, 40% and 60%, 60% and 90%, and values >90% will be interpreted as low, moderate, substantial, and considerable heterogeneity, respectively [108]. The I^2^ test is considered to be a more reliable option when the number of studies included in the meta-analysis is small [108,109].


Assessment of Publication Bias


Publication biases and small-study effects will be assessed using funnel plots, as well as Egger’s and Begg’s tests [110,111]. Funnel plots asymmetry and *p*-values below 0.10 for Egger’s and Begg’s test will be considered as indicative of potential publication biases [110,111].


Certainty of Evidence


The certainty of evidence will be assessed using the Grading of Recommendations, Assessment, Development and Evaluations (GRADE) approach [112]. Summary of finding tables will be created for each dietary variable using the GRADE profiler Guideline Development Tool [113]. Two authors (A.B. and C.V.) will independently assess the certainty of evidence using the five GRADE domains, namely risk of bias, inconsistency of results, indirectness of evidence, imprecision of results, and publication bias. Disagreement between authors performing the GRADE assessment data extraction will be solved through discussion and consensus involving all authors. As recommended by the GRADE Working Group, the certainty of evidence for each dietary variable and the outcome will be rated as:(i).*High certainty:* We are very confident that the true effect lies close to that of the estimate of the effect.(ii).*Moderate certainty:* We are moderately confident in the effect estimate. The true effect is likely to be close to the estimate of the effect, but there is a possibility that it is substantially different.(iii).*Low certainty:* We have little confidence in the effect estimate. The true effect may be substantially different from the estimate of the effect.(iv).*Very low certainty:* We have very little confidence in the effect estimate. The true effect is likely to be substantially different from the estimate of effect.

Any downgrades in the certainty of evidence will be justified using footnotes.

## 3. Discussion

The proposed review will provide a rigorous and comprehensive synthesis of existing evidence on the association between a wide range of dietary variables and depression. Beyond summarizing the findings for a wide range of dietary variables, updated meta-analyses along with subgroup and sensitivity analyses will be performed. As results of each primary study included in selected systematic reviews and meta-analyses will be extracted, the review will help address the problem of overlapping systematic reviews and conflicting findings. It will further help to specify the common sources of bias in studies in this field as well as the needs in terms of future research. Potential limitations of the proposed methods include heterogeneity issues, and a higher risk of bias due to publication language restriction as well as the fact that we will not seek unpublished reports.

Upon completion, this review will be the most comprehensive and up-to-date synthesis of currently available evidence on the relationships between diet and depression. It will serve as a key reference for guiding future research and as a resource for health professionals in the fields of nutrition and psychiatry.

## Figures and Tables

**Figure 1 mps-06-00078-f001:**
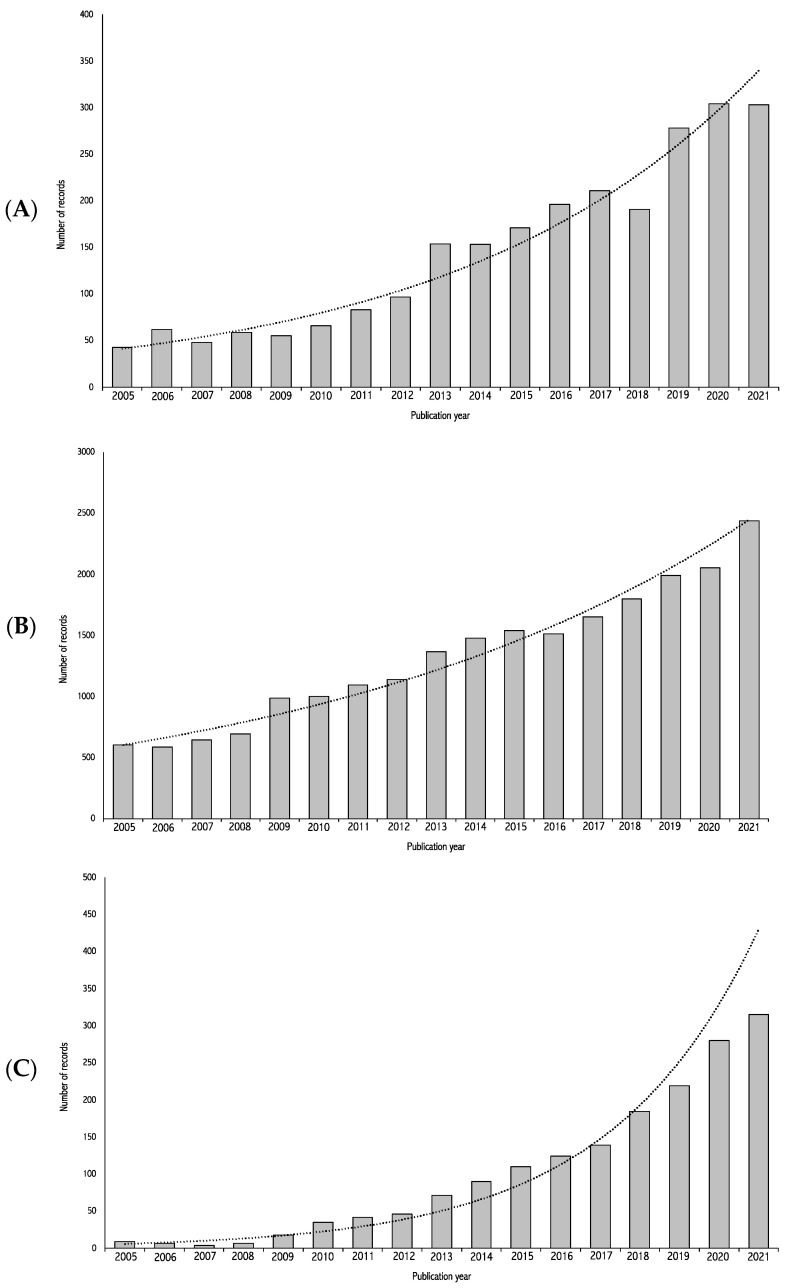
Number of non-duplicate records of (**A**) systematic reviews and meta-analyses on diet and depression, (**B**) any type of publication on diet and depression and (**C**) any type of publication on dietary patterns and depression published between 2005 and 2020. Search details graphic (**A**): diet.mp. AND depression.mp. AND ((systematic ADJ1 review) OR (meta-analys*).mp.) searched in Medline (Ovid), EMBASE (Ovid), PsycINFO (Ovid), Cochrane Database of Systematic Reviews. Duplicates were removed. Search details graphic (**B**): diet.mp. AND depression.mp. searched in Medline (Ovid), EMBASE (Ovid), PsycINFO (Ovid), and Cochrane Central Register of Clinical Trials. Duplicates were removed. Search details graphic (**C**): (dietary ADJ1 pattern*).mp. AND depression.mp. searched in Medline (Ovid), EMBASE (Ovid), PsycINFO (Ovid), and Cochrane Central Register of Clinical Trials. Duplicates were removed.

**Figure 2 mps-06-00078-f002:**
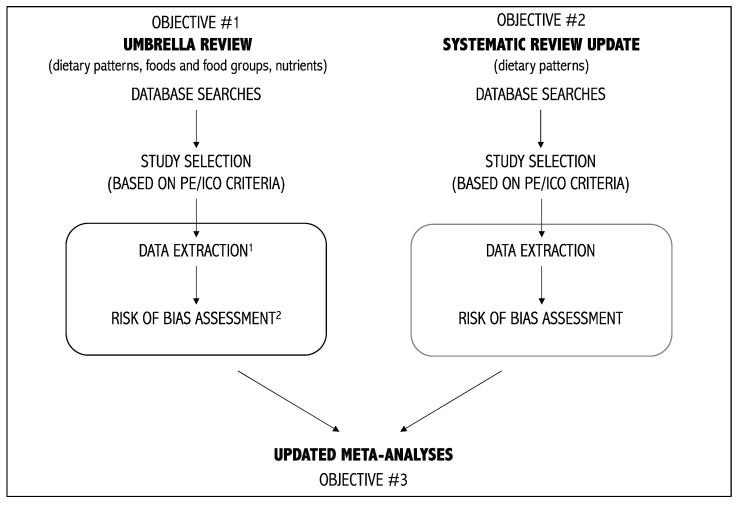
Schematic representation of the proposed work. ^1^ We will perform extraction of (1) systematic reviews’ characteristics and results and of (2) primary studies’ characteristics and results. ^2^ We will assess (1) the methodological quality of systematic reviews and the risk of bias in primary studies. P: Population(s); E: Exposure(s); I: Intervention(s); C: Comparator(s); O: Outcome(s).

**Table 1 mps-06-00078-t001:** PsycINFO (through Ovid) search strategy for identifying systematic reviews on diet and depression.

	Search Terms
Theme #1 Diet	diets/ food intake/ eating behaviour/ exp appetite/ diet *.mp. nutriti *.mp. food.mp. eat *.mp. (energy intake).mp. macronutrient *.mp. micronutrien *.mp nutrient *.mp. 1 OR 2 OR 3 OR 4 OR 5 OR 6 OR 7 OR 9 OR 10 OR 11 OR 12
Theme #2 Depression	14. major depression/ 15. dysthymic disorder/ 16. recurrent depression/ 17. treatment resistant depression/ 18. depression *.mp. 19. (depressive ADJ3 (condition * OR disorder * OR symptom *)) 20. 14 OR 15 OR 16 OR 17 OR 18 OR 19
Type of studies	21. systematic review/ 22. meta-analysis/ 23. (systematic ADJ3 review).mp. 24. meta-analysis.mp. 25. (meta ADJ2 analysis).mp. 26. 21 OR 22 OR 23 OR 24 OR 25
Combining search terms	27. 13 AND 20 AND 26
Language	28. limit 27 to (english or french)
Search period	29. limit 28 to yr = “2005-current”

**Table 2 mps-06-00078-t002:** Data extraction fields for systematic reviews.

	Extraction Fields
General information	Authors Publication year Type of review (with/without meta-analysis) Protocol registration (yes/no) Protocol registration number (where applicable) Protocol publication (yes/no) Protocol publication reference (where applicable)
Search strategy	Number of databases searched Names of databases (via platform) searched Date search beginning Date of last search Grey literature searched (yes/no) Names of grey literature searched Other types of searches (where applicable)
Study selection	Number of studies included in the systematic review Number of studies included in the meta-analysis (where applicable) Number of studies included in the systematic review by study design
Population	Sex Age(s) and life stage(s) Statistical subgroup analysis by sex (yes/no)
Exposure(s) Intervention(s)	Description of the dietary variable(s) or in intervention(s) Tool(s) used to assess dietary variable(s) or adherence to intervention Variable(s) type (dichotomous/nominal/ordinal/continuous)
Comparators	Descript on of the dietary intervention(s) Tool(s) used to assess dietary variable(s) or adherence to intervention
Outcome(s)	Description of the outcome(s) Tool(s) used to assess the outcome(s) Variable(s) type (dichotomous/nominal/ordinal/continuous)
Results	Summary of qualitative results Quantitative results with description of statistical analysis type Authors’ conclusions

**Table 3 mps-06-00078-t003:** Data extraction fields for primary studies.

	Extraction Fields
General information	Authors Publication year Protocol registration number (where applicable) Protocol publication reference (where applicable)
Study charactaristic	Study location Study design Study duration (where applicable) Participants recruitment type Type of randomization (where applicable)
Population	Sex Number of participants Number of participants by sex Age(s) and life stage(s) Statistical subgroup analysis by sex (yes/no)
Exposure(s) Intervention(s)	Description of the dietary variable(s) or in intervention(s) Tool(s) used to assess dietary variable(s) or adherence to intervention Variable(s) type (dichotomous/nominal/ordinal/continuous)
Comparator(s)	Description of the dietary intervention(s) Tool(s) used to assess dietary variable(s) or adherence to intervention
Outcome(s)	Description of the outcome(s) Tool(s) used to assess the outcome(s) Variable(s) type (dichotomous/nominal/ordinal/continuous)
Results	Quantitative results with description of statistical analysis type Quantitative results by sex with description of statistical analysis type Adjustments (where applicable)

**Table 4 mps-06-00078-t004:** Criteria for judging the overall quality of systematic reviews and meta-analyses using the Assessment of Multiple Systematic Reviews 2 (AMSTAR-2) tool.

		Criteria for Judging the Overall Quality of SR			
Quality	Level of Confidence in the Results	Critical Weaknesses ^1^ (N=)	Non-Critical Weaknesses ^2^ (N=)	Score ^3^ Cut-Offs for Downgrading Due to Multiple Non-Critical Weaknesses	
				SR without MA (__/13)	SR with MA (__/16)
High	Provides an accurate and comprehensive summary of available studies.	0	1	-	-
Moderate	May provide an accurate and comprehensive summary of available studies	0	≥2	<8	<11
Low	May not provide an accurate and comprehensive summary of available studies	1	0 or ≥1	<8	<9.5
Critically low	Does not provide an accurate and comprehensive summary of available studies	≥2	0 or ≥1	-	-

AMSTAR-2: Assessment of Multiple Systematic Reviews 2; MA: Meta-analysis(es); SR: Systematic review(s). ^1^ Systematic reviews without meta-analysis(es)): “No” answers to items 2, 4, 7, 9, and 13 will be considered critical weaknesses. Systematic reviews with meta-analysis(es)): “No” answers to items 2, 4, 7, 9, 11, 13, and 15 will be considered critical weaknesses. ^2^ Systematic reviews without meta-analysis(es)): “Partial yes” answers to items 2, 4, 7, and 9 and “no” answers to items 1, 3, 5, 6,8, 10, 14, and 16 will be considered non-critical weaknesses. Systematic reviews with meta-analysis(es)): “Partial yes answers” to items 2, 4, 7, and 9 and “no” answers to items 1, 3, 5, 6,8, 10, 12, 14, and 16 will be considered non-critical weaknesses. ^3^ A total score will be calculated, with each “no”, “partially yes”, and “yes” answer being attributed 0, 0.5 and 1 points, respectively. As items 11, 12 and 15 are only applicable to systematic reviews including meta-analyses, systematic reviews without meta-analysis(es) have a maximum possible score of 13 and systematic reviews with meta-analysis(es) have a maximum possible score of 16. For systematic reviews judged as being of moderate quality based exclusively on the number of critical and non-critical weaknesses, scores < 8 (systematic reviews without meta-analysis(es)) and <11 (systematic reviews with meta-analysis(es)) indicate that they have ≥5 non-critical weaknesses [92].

**Table 5 mps-06-00078-t005:** PsycINFO (through Ovid) search strategy for identifying primary studies on dietary patterns and depression published outside the timeframe of previous systematic reviews.

	Search Terms
Theme #1 Diet	diets/ diet.mp. exp appetite/ (diet * pattern *).mp. (diet * quality *).mp. (eating pattern *).mp. (food pattern *).mp. 1 OR 2 OR 3 OR 4 OR 5 OR 6
Theme #2 Depression	8. major depression/ 9. dysthymic disorder/ 10. recurrent depression/ 11. treatment resistant depression/ 12. depression *.mp. 13. (depressive ADJ3 (condition * OR disorder * OR symptom *)) 14. 8 OR 9 OR 10 OR 11 OR 12 OR 13
Combining search terms	15. 7 AND 14
Language	16. limit 15 to (english or french)
Search period	17. limit 16 to yr = “2005-current”

## Data Availability

Not applicable.

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
