# Peer review of "Gathering the Evidence on Diet and Depression: A Protocol for an Umbrella Review and Updated Meta-Analyses"

_mps, 2023, doi:10.3390/mps6050078_

Round 1

Reviewer 1 Report

The revised manuscript describes the relevance of the study topic and details the methodology to be followed in the systematic review. But, in order to make the text more understandable and less repetitive, it is suggested to the authors:

1. Rephrase the objectives more clearly: the three objectives are cited in sing numbered bullet points, and then specific those. It is suggested to explain and specify each objective at the same time (for example: 1 an umbrella review on diet and depression. This objective will allow us to synthesize the characteristics, methodology, results, and extent of overlap of existing systematic reviews on diet and depression. 2.)

2. The methodology has too many sections. It is recommended to reorganize the information explaining the methodology to be followed for each of the objectives on a continuous basis. In the current format, this section is divided into different items, mixing all the objectives in them. This causes content to be repeated and the thread of the proposed methodology for each objective to be lost.

3. The discussion section has content already explained in the introduction. It is recommended to replace this section, for example, by expected results / contribution of the study.

4. Please, replace the word “race” (line 141) with a more appropriate one. We should not use the word race when talking about human populations.

Reviewer 2 Report

- Please add "depression" and "anxiety in the Keywords. 

- Please add the following publications in the introduction (Line 67: the association between diet and depression):

https://pubmed.ncbi.nlm.nih.gov/32885996/

- Authors must determine if they will pool OR, RR and HR for observational studies or they will consider these values separately. (Line 329) 

Reviewer 3 Report

Dear Authors, 

The manuscript addresses a very helpful protocol for conducting a systematic review related to depression and dietary patterns. This manuscript is not ready for publication yet. There are some minor issues that the authors need to modify before the manuscript can be considered for publication. The following are my comments:

1. Reduce number of citing references.  The references are around 88 in introduction.  

2. Methods are so long. Please summarize methods. 

3. In table 1, can you insert some terms more related o dietary pattern like Western dietary pattern, healthy dietary pattern, vegetarian, Mediterranean diet, etc?
